# Mechanistic Modelling of Radiation Responses

**DOI:** 10.3390/cancers11020205

**Published:** 2019-02-10

**Authors:** Stephen J. McMahon, Kevin M. Prise

**Affiliations:** Centre for Cancer Research & Cell Biology, Queen’s University Belfast, Belfast BT9 7AE, Northern Ireland, UK; k.prise@qub.ac.uk

**Keywords:** radiation, radiobiology, mathematical modelling, mechanistic modelling, track structure, DNA repair, cell death

## Abstract

Radiobiological modelling has been a key part of radiation biology and therapy for many decades, and many aspects of clinical practice are guided by tools such as the linear-quadratic model. However, most of the models in regular clinical use are abstract and empirical, and do not provide significant scope for mechanistic interpretation or making predictions in novel cell lines or therapies. In this review, we will discuss the key areas of ongoing mechanistic research in radiation biology, including physical, chemical, and biological steps, and review a range of mechanistic modelling approaches which are being applied in each area, highlighting the possible opportunities and challenges presented by these techniques.

## 1. Introduction

In contrast to many other areas of biology, mathematical modelling has been an essential part of the fields of radiation biology and radiation therapy since their inception. Close links between the disciplines of physics, biology, and medicine have underpinned this connection, fostering a robust field of physical and biological modelling of radiation responses. On the side of physics, advanced mathematical methods underpin the optimization of modern radiotherapy techniques, enabling precise conformation of dose to target tumors. However, mathematical modelling can also contribute to advancing our understanding of fundamental radiobiology, and has done so since the very earliest studies.

Mathematical modelling has been applied in radiobiology since the 1920s, in studies of proliferation in bacteria and chick embryos [1,2]. In both cases, an exponential dependence of cell viability on radiation was obtained, leading to the development of a “target theory” hypothesis of radiation response. This suggested that cells contain a sensitive region (or “target”) which could be inactivated by a damaging event (or “hit”), and that the yield of these hits is proportional to the dose delivered. If radiation-induced is induced randomly among cells, these the number of hits per cell will follow a Poisson distribution, with a mean number per cell of $n=D/D_{0}$, where D is the dose delivered and $D_{0}$ is the dose which causes one “hit” on average. If only un-hit cells survive, then the survival probability is the same as the probability of a cell having 0 hits, that is: $S=P_{hits}\left(0,D\right)=e^{-\frac{D}{D_{0}}}$, where $P_{hits}\left(0,D\right)$ is the probability of 0 hits following exposure to a dose D. An exponential dependence results, as the probability of seeing 0 hits in a Poisson distribution with mean n is given by $e^{-n}.$

These models proved highly effective in studies on bacteria and other simple systems, but evidence grew that not all response curves showed this simple exponential dependence [3], with some systems showing evidence of a “shouldered” response curve, with initially lower sensitivity to radiation. Interest in this type of model expanded rapidly after the development of robust techniques to assess the in vitro radiation sensitivity of mammalian cells [4,5], which consistently showed shouldered response curves.

Multi-target models sought to address this by suggesting that rather than a single target, cells contained multiple targets, all of which must be inactivated to cause cell death. If each target still had the same $e^{-\frac{D}{D_{0}}}$ hit probability, then the probability of a cell surviving is $S=1-{\left(1-e^{-\frac{D}{D_{0}}}\right)}^{n}$, where n is the number of targets in the cell. This gave a survival curve with an initial slope of 0, which gradually increased to $1/D_{0}$ at high doses (on a log scale). While the multi-target model was able to successfully fit many observed cell-survival datasets, questions remained about its wider applicability. In particular, a growing body of data indicated that many cell lines did not have zero initial slope, and experiments at high doses often showed increasing curvature, up to the limit of detectability [6,7]. 

One alternative survival model, independently proposed by a number of authors, was the linear-quadratic model $S=e^{-\alpha D-\beta D^{2}}$, which incorporated both a linear initial slope (governed by α) and a continually increasing quadratic curvature (governed by β). The key features of this Linear-Quadratic (LQ) model are illustrated in Figure 1, showing the contributions of the linear and quadratic response components. The LQ has become the dominant mathematical model in modelling the survival of cells in preclinical studies, in part motivated by its close links to clinical observations. Studies of fractionation effects in tissues and tumors demonstrated that not only did the LQ model effectively reproduce in vitro survival, but also the effect of clinical fractionation, providing a consistent way to interpret these effects [8,9,10]. 

One important feature of the LQ is that despite its relative simplicity, it provides an intuitive explanation for a range of biological effects. For example, the tissue-sparing benefits of fractionation can be understood by noting that if cells are allowed to repair between two fractions of radiation, they repair “sub-lethal” damage and repeat the shouldered portion of the curve, seeing less killing than if the dose is delivered in a single fraction. The degree of sparing depends on the degree of curvature, typically quantified in terms of the α/β ratio. This has units of Gy, and corresponds to the dose at which the linear and quadratic components have equal contributions to cell killing. Cells with a low α/β ratio see superior sparing compared to those with a high α/β ratio, as illustrated in Figure 1. As a result, the LQ remains the foundation for determining the equivalence of different treatment fractionation schedules in the clinic [11]. 

However, the LQ is not without its challenges, and there remain outstanding questions about its interpretation. In addition to its applications in the analysis of clonogenic survival and fractionation, quadratic-type responses had also previously been fit to other endpoints, such as the formation of chromosome aberrations and mutations [12,13]. In each case, the general features driving the relationship are broadly similar; for example, that the linear component relates to “single-hit” damage resulting from individual incident particles, and the quadratic component rates to the strength of interaction of accumulated radiation effects from multiple incident particles. However, the exact mechanisms underlying these trends were the subject of some debate.

As the LQ gained increased prominence as a model of survival in the 60s and 70s, a range of quantitative models were developed to attempt to understand these trends. In 1972, Kellerer and Rossi published proposed that the yield of “elementary lesions” followed a quadratic dependence with, dose being formed either directly by a single event, or as “dual lesions” due to the interaction of two sub-lesion events, in the “theory of dual radiation interaction” [14]. In this model, the nature of the “lesions” were deliberately left unspecified. However, the following year Chadwick and Leenhouts published their “molecular theory of cell survival”, which took a similar approach to predicting the sensitivity of cell survival, but explicitly identified the critical lesions as DNA double strand breaks, and sub-lesions as single strand breaks [15]. Although the two models had somewhat different underlying assumptions, in both cases they could reproduce LQ trends at low doses, in good correspondence with experimental observations. However, this was not the only approach taken.

An alternative family of models considered a linear yield of initial lesions, and modeled the quadratic term of radiation response as being due to lesion interaction. Among the first comprehensive model was the Repair-Misrepair (RMR) model of Tobias et al. [16,17]. In this model lesions are induced linearly with dose, and can be repaired with either linear kinetics, or quadratic kinetics proportional to the density of lesions. The former process can be interpreted as correct end joining, and the latter as binary misrepair. Each of these types of joining can be associated with a probability of lethality, giving rise to response curves which can replicate a number of different response models, including target theory models and the LQ. 

A number of other models have been developed using similar concepts of linear and quadratic repair processes, with some minor variants on approach. Another extensively studied approach is the Lethal-Potentially Lethal (LPL) model of Curtis [18]. In this model, rather than a single class of lesion, there are “lethal” lesions, which are always fatal to the cell, and “potentially lethal” lesions, which are repaired with linear and quadratic kinetics, as in the RMR. The linear component is treated as correct rejoining, while the quadratic component is taken as incorrect rejoining and also considered to be lethal. 

One final class of model, rather than explicitly postulating direct lesion interaction as the driver of misrepair, suggested that cells have a finite repair capacity. One such Saturable Repair model was proposed by Goodhead [19], which suggested that as the number of lesions increased, the probability of successful repair fell, due to a reduction in the cell’s ability to repair all of the induced damage, for example, due to enzyme or energy depletion. As this means additional dose causes increasing lethality, a shouldered response curve can be formed, potentially closely mimicking the LQ model for suitable parameters. 

All of these models have several desirable features—they provide a mechanistic link between the LQ and fundamental damage and repair processes, and can potentially be applied to predict yields of other endpoints, such as mutations, as well as the cell-protecting effects of fractionation or the Relative Biological Effectiveness (RBE) of charged particles (discussed in more detail below). 

Most of these models are sufficiently general to be applied in a range of scenarios. However, in the majority of cases, the limitations of experimental clonogenic survival data mean that clinical results cannot adequately distinguish between these models, as they can all predict broadly similar dose response curves with suitable parameter selection (reviewed in [20]). It has thus proven challenging to robustly demonstrate which, if any, of these models truly describe the mechanisms underpinning the LQ.

This presents difficulties for the translation of the LQ into modern radiotherapy. The benefits of fractionation are typically explained in terms of the 4 R’s of radiotherapy—repair, reoxygenation, redistribution, and repopulation [21]—and while some of those (most notably repair) are naturally incorporated in the LQ model, others factors are more difficult to incorporate without the inclusion of further empirical fitting parameters. Moreover, there is a growing interest in a range of other factors which alter radiation responses, including radiations of different qualities [22], radiation-response modifying drugs [23], microenvironmental features of the tumor [24], and inter-patient variations in radiation sensitivity [25]. 

In addition, as the LQ model is applied as an empirical fit to survival, the parameters obtained for a given cell or tissue do not provide any information about other potentially biologically relevant endpoints of radiation exposure, such as the formation of potentially carcinogenic mutations. These limitations, coupled with the burgeoning knowledge of the mechanisms underlying radiation response, have driven considerable interest in the development of more mechanistic models of radiation response and cellular fate. 

However, as illustrated in Figure 2, radiation damage is initiated by physical interactions on the timescale of femtoseconds, triggering a cascade of processes that can result in alterations to the cell’s fate hours, days, or even years following the initial exposure, mediated by both chemical and biological processes. While there are many models of both individual processes and combinations relevant to specific endpoints, because of the range of processes and scales involved, there is not as yet a single integrated model of all of these processes. In this review, we will discuss the modelling efforts on these different scales, which attempt to better understand cellular responses to ionizing radiation, and briefly discuss how this may build towards improved individualized predictions.

## 2. Physical DNA Damage

The genetic content of the cell was putatively identified as the primary target for radiation action some years before the structure of DNA was determined [13], a prediction which has been extensively validated in a range of subsequent studies [26,27,28]. In particular, it has been shown that DNA double strand breaks (DSBs)—events which sever both strands of the DNA in close proximity—are strongly correlated with lethality. As discussed in more detail in Section 3, this is because although cells have a number of pathways which can effectively repair DSBs, these are more error-prone than those which repair simpler events, such as base damages or single-strand breaks [29]. 

On the macroscopic scale, radiation dose is frequently quoted as a spatially averaged quantity across relatively large volumes—whole cell populations or tumors—but this does not fully reflect the complexity of radiation damage. DNA has structure on the nanometre scale, with a double-helix having a diameter of approximately 2 nm, and DSBs involving breaks separated by approximately two turns or fewer, corresponding to on the order of 10 to 20 base pairs, or 4–8 nm in length. As a result, physical differences in energy deposition on this scale lead to differences in biological effect. While “sparsely ionizing” radiation, such as high energy X-rays, distributes its energy effectively randomly around the DNA, high LET radiation (such as heavy charged particles) deposit their energy much more densely. This leads to damage with greater complexity, potentially in greater amounts per unit dose, which the cell finds more difficult to repair (schematically illustrated in Figure 3).

This elevation of damage means that densely ionizing particles exhibit a high Relative Biological Effectiveness (RBE). This is defined as the ratio between the dose of a sparsely ionizing reference radiation which gives rise to a particular biological effect, and the dose of a densely ionizing radiation which has the same biological effect. The exact magnitude of RBE and its dependence on parameters such as dose and LET is of particular concern in the delivery of radiotherapy with charged particles, such as protons and ions, making this a topic of considerable research interest in radiobiology. 

While many techniques are available to model energy depositions on the nanoscale, the “gold standard” is provided by Monte Carlo track structure calculations [30]. Such simulations stochastically model radiation interactions on an event-by-event basis, tracking each scattering, excitation, or ionization event of both the primary and any secondary particles which are produced (giving “tracks” as illustrated in Figure 3). Such models have been in use for more than three decades. Early research focused on microdosimetric energy distributions, assessing the stochastic distribution of energy on the sub-cellular scale [31,32]. This work demonstrated the high degree of heterogeneity which can be seen in the distribution of energy even within cells which are exposed to the same average dose, highlighting the need for more detailed simulations to accurately predict radiation damage following ionizing radiation exposure.

In the early 1990s, significant developments were made advancing this approach from simple microdosimetric calculations to models incorporating key features of DNA structure. This enabled calculations of energy depositions in various sub-regions of DNA, and predictions of the damage to the DNA structure which would result [33,34]. By fitting the sensitivity of various parts of the DNA to ionizing events, such as the amount of energy which must be deposited in a sugar-phosphate backbone to cause a strand break, predicted yields of measurable endpoints (such as SSB and DSB) could be tuned to match experimentally observed values, and the models used to interrogate underlying mechanisms and endpoints which were not so readily measurable, such as the yield of complex, multiply-damaged sites [26].

Historically, these Monte Carlo codes were typically developed ad-hoc by individual groups for their particular research purposes, and so were frequently restricted in terms of their energy range and simulated particles (reviewed in [35]). A wide range of codes have been applied in simulations of dosimetry on both the patient and cellular scale, including MCNP [36], EGS [37], FLUKA [38], KURBUC [39], MC4 [40], RITRACK [41], TRAX [42], PHITS [43], PARTRAC [44], and Geant4 [45]. Many Monte Carlo codes make use of a “condensed history” physics model, where some charged particle interactions are condensed into single steps to improve performance (including MCNP, EGS, FLUKA, Geant4). While this accurately reproduces results on the patient scale, it is insufficient to fully explain interactions on the cellular scale. Instead, track-structure models which simulate each interaction individually are needed to fully describe the interaction of radiation on the cellular scale [46,47].

As a result, a number of dedicated codes have been developed for these purposes (including KURBUC, MC4, RITRACK, TRAX, PHITS, and PARTRAC), and general-purpose Monte Carlo codes have been extended to calculate track structure at lower energies, making them applicable to cellular scales. Probably the most widely used code in this area is Geant4, a general-purpose Monte Carlo toolkit originally designed for high-energy charged particle transport, which has been extended to encompass ionizing events at energies relevant to cellular nanodosimetry (down to a few eV per particle) through the Geant4-DNA project [48,49]. Geant4 also underpins the TOPAS and TOPAS-nBio toolkits [50,51], which seek to make these Monte Carlo approaches more generally accessible.

These different codes have seen a range of levels of uptake in different fields and applications, but although all of these codes are capable of simulating nanoscale track structures, there remain a number of outstanding research areas where further development is needed to more robustly predict DNA damage from first principles. These include cross section models, descriptions of radiation chemistry, DNA structures and geometry, and understanding of how these different events combine to give rise to DSBs.

Interaction cross-sections are one of the key pieces of input data needed for Monte Carlo models, describing the probability of various scattering and ionization events as a function of energy and target material. However, the majority of nanoscale Monte Carlo calculations simply treat all organic material as water-equivalent. While this is a good approximation at high energies, at low energies these assumptions may break down [52,53]. As a result, there is ongoing research to attempt to measure and model relevant cross-sections in organic molecules, including DNA, so that they can be more robustly incorporated in the Monte Carlo models to provide better predictions of the damage ionizing radiation may have on DNA.

Radiation chemistry is also an area of substantial development. It is well-known that direct interactions between the ionizing particles and DNA are responsible for only approximately 30% of initial damage for photon irradiation, and a somewhat greater proportion for heavier charged particles. Instead, so-called “indirect” interactions play a dominant role. These are events where radiation interacts with another molecule (predominantly water), creating ions or radical species which proceed to react with the DNA, damaging its structure and leading to strand breaks. While early modelling simulated these radical species in small test volumes [33], simulating such reactions in realistic cell volumes can be extremely time-consuming, because of the large number of species which can be generated. Codes are under development, which provide this functionality more generally, including as part of PARTRAC [44], Geant4-DNA [54], TOPAS-nBio [55], and TRAX-CHEM [56]. However, in most cases these codes are still limited to simulations in liquid water, due to the lack of information on reaction rates in more complex chemical compositions.

Because of the above approximations, actual DNA structures are also frequently neglected. Instead, homogeneous water volumes are simulated, with DNA structures superimposed on the simulation results to determine the distribution of damaging events. While this avoids complications from the lack of cross-sections for relevant targets, the appropriate DNA structure to use remains an outstanding research question. Early work [34] made use of small sections of DNA (equivalent to single DNA strands of tens of base pairs in length), which is sufficient to calculate rates of local damage in an “isolated” DNA section. However, DNA has a number of higher-degree levels of organization, with individual strands wrapped around histones to form nucleosomes, which are in turn packed into larger fibers. This organization means that DNA damage is not randomly distributed, but may depend on local structure around a radiation track. A range of studies have modelled different DNA structures [44,57,58,59], and shown that the choice of a particular DNA structure can have substantial impacts on the predicted yields of different types of damage, and how it is distributed through the genome. 

Despite the above uncertainties, numerous models have been generated which predict the yield of radiation-induced DSBs in a variety of systems, based on models of either direct damage alone, or combinations of direct and indirect damage. These range from multi-scale models which simulate damage in detail on small DNA structures and extrapolate to averaged or cell-level responses [58,60,61,62]. There are also a range of whole-cell models incorporating varying levels of detail on DNA damage and chromosome structure, summarized in Table 1. Although these models have differences in their design, underlying physics, whether or not they include chemical effects, and models of biological structure and DNA damage, many of them produce broadly similar predictions.

In part, this is the result of the above-mentioned significant uncertainty in the nature of the physico-chemical processes, which give rise to double-strand breaks at the interface between physical and biological models. Due to the lack of distinction between different modelling assumptions, there is very large uncertainty in the underlying biological parameters, and even quite different models can be brought into a reasonable agreement through careful selection of different parameters. These include the amount of energy which needs to be deposited in a single strand to cause a break (with threshold energies estimated from a few eV up to 40 eV, or in some cases an energy-dependent probability [34,61,67]), the likelihood of radical interaction causing damage, and how different damages in close proximity interact. This problem is compounded by the often significant uncertainties present in experimental measurements of DNA damage, which make determining absolute yields of many types of damage challenging. 

Thus, while a range of models exist to predict the yield and distribution of DNA damage for different types of irradiation, there is a significant need to better characterize many of the fundamental parameters governing these models, to refine their underlying structure, and enable more robust comparisons between models. A recent collaboration has proposed a Standard for DNA Damage reporting, enabling the outputs of various models to be efficiently compared, to help facilitate such developments [68].

## 3. DNA Repair

Although DNA damage is the primary driver of radiation-induced effects, it represents only the initiating event in a cascade of processes. Living systems have evolved a wide range of systems enabling the efficient repair of DNA damage, so only a very small fraction of even double-strand breaks lead to lethality. While radiation induces 30–40 DSBs per cell per Gy, this typically corresponds to less than one lethal event per cell [69]. Failures in these repair processes lead to lethality, either through the failure to repair a subset of DSBs [70,71] or misrepair events, which cause significant chromosomal abnormalities [72]. This is particularly apparent in cell lines with defects in key DNA repair genes, which can be more than an order of magnitude more sensitive than repair-competent cells.

Due to their lethality, the repair of DSBs has been the subject of the majority of the modelling efforts in this area. A number of high-level models of DNA damage were developed to attempt to understand how damage may combine into different types of damage. A number of similar approaches considered proximity effects as the major driver of misrepair, with spatially dense clusters of damage having a high probability of incorrect end joining, leading to potentially deleterious misrepair [73,74,75,76,77,78]. Such models often proved effective at predicting yields of events, such as mutations [76] and chromosome aberrations, which can in turn be linked to cell death in some cases [76,79]. However, most such models do not incorporate detailed models of the underlying repair processes, which limits their capacity to incorporate our growing mechanistic understanding of DNA repair.

As schematically illustrated in Figure 4, there are three primary mechanisms by which DSBs can be repaired: Nonhomologous End Joining (NHEJ), Homologous Recombination (HR), or Microhomology-Mediated End Joining (MMEJ). NHEJ is active throughout the cell cycle, and rapidly repairs most DNA DSBs, although is somewhat error-prone and often leads to small modifications around the site of repair [80]. By contrast, HR makes use of DNA from a matching chromosome to precisely repair DNA without introducing new sequence errors. However, because of the dependence on a template strand, it is limited to activity in late S and G2 phases, where DNA has been replicated [81]. Finally, as a backup to these processes, MMEJ can act to rejoin DNA. In this process, DNA strands are matched based on small regions of homology, typically involving significant resection at break ends, and is very error-prone and almost always leads to at least moderate sequence alteration [82]. A number of potential pathways have been implicated in MMEJ, including a resection-dependent subset of NHEJ [83], as well as a wholly distinct pathway known as alternative-NHEJ, which is independent of most major NHEJ proteins [84].

The kinetics of these different repair pathways has been the subject of a range of experimental and modelling studies. The most common empirical approach describes DNA repair as a single- or multi-exponential process, with different sub-populations of breaks being repaired with different kinetics. However, other explanations for the slowing rate of repair at later times have also been suggested, such as a second-order exponential dependence [85] or more complex repair interplays with multiple kinetic parameters or variable repair-times (summarized in [86]). Some of these models, such as the multi-exponential approach, offer a seemingly natural link between groups of breaks repaired with different kinetics and the various available pathways, although there is evidence that other features of breaks, such as their complexity and the local structure of DNA, may also impact on repair kinetics. 

Some models of DNA repair fidelity and survival have sought to incorporate detail on these pathways, reflecting them as different repair processes which still act in a largely probabilistic fashion [76]. While still relatively abstract, such models still enable predictions of differential responses in cells with different genetic backgrounds, building towards individualization of radiation response predictions.

Much more detailed studies seek to fully model the biochemistry of these pathways, tracking the migration and interaction of DNA ends and proteins, either using population biokinetics and probabilistic models [87,88,89] or by modelling individual agents in a Monte Carlo fashion [90,91]. These models have primarily focused on NHEJ, and have demonstrated the ability to reproduce observed trends in DNA repair kinetics and the impact of knocking out some key genes [92], potentially providing a framework to better understand these processes on a quantitative level.

By contrast with NHEJ, no quantitative mechanistic models of the other major DNA repair pathways, HR and MMEJ, have been published in combination with radiation. For HR, this is in part due to its complexity, as it involves template DNA strands and a range of potential mechanisms for the resolution of the resulting D-loop (or Holliday Junction) formed between the repaired and template chromatids, the relative activity of which has yet to be fully determined [93]. In the case of alternative-NHEJ, the pathway has only been recently identified as distinct from NHEJ, and while many of the key genes involved have been identified [94], there has been relatively limited modelling of their function.

To fully model the consequences of ionizing radiation, robust descriptions of these pathways are needed to accurately predict the consequences of repair. This includes not only the activity and fidelity of individual pathways, but also how they interact with one another to determine which pathway repairs a particular break, and how this depends on the physical complexity of damage and the potential presence of other damage within the nucleus.

At present, there are a number of models, based on both simple proximity-based or more sophisticated molecular kinetics, which make predictions that can potentially be experimentally tested, relating to yields of different types of misrepair under different irradiation conditions. However, as with initial physical damage, the extensive parameter space underpinning these responses make robustly fitting and falsifying different models challenging—even relatively simple kinetic models of NHEJ can involve between 10 and 20 fitting parameters [88,92].

## 4. Cell Fate

Once initial DNA damage distributions and their repair have been determined, there is still the question about the cell’s eventual fate. A wide range of approaches has been taken to incorporate our knowledge of these underlying processes into radiation response models.

As noted above, many early radiation response models involved simplified descriptions of how radiation response processes were believed to proceed. Approaches like the dual action model of Rossi and Kellerer [14], the Repair-Misrepair model of Tobias [16], the lethal-potentially lethal model of Curtis [18], or the molecular theory of Chadwick and Leenhouts [15] represented some of the first efforts of incorporating knowledge of radiation biology into mechanistic response models. Such approaches proved successful at capturing the basic features of the radiation dose response curve, and there are a number of modern developments which have sought to place these on a more biologically robust footing by incorporating additional detail about the underlying biology [89,95,96,97]. However, a range of other approaches also seek to build on these more general assumptions to incorporate further mechanistic details about the underlying radiation physics and biology. 

For example, predicting the RBE of high LET radiations has been a topic of considerable interest, due to both the interest in the underlying biology as well as the implications for clinical practice in facilities which seek to use heavy charged particles for the treatment of cancer. Many empirical models have been developed which predict protons RBE-LET dependence by applying simple modifications to LQ parameters, typically of the form $\alpha_{p}=p_{1}\alpha_{x}+\frac{p_{2}LET}{\alpha_{x}/\beta_{x}}$, where $\alpha_{x}$ is the α LQ parameter for protons or X-rays respectively, $p_{1}$ and $p_{2}$ are empirical fitting parameters, and $\alpha_{x}/\beta_{x}$ is the ratio for X-ray responses [98,99,100]. Similar modifications are sometimes also proposed for the β term. While broadly successful for proton therapy, such approaches are typically ineffective at the higher LETs used in therapy with heavier ions, such as carbon.

A number of mechanistic explanations of cell inactivation were explored to attempt to better understand the mechanisms underpinning these effects. This includes early work, such as geometric radial energy distribution models developed by Butts and Katz [101] and applications of the dual action model [14], as well as more recent modelling approaches, including the Repair-Misrepair-Fixation model (RMF) [89] and many extensions of the DNA damage models, outlined in Section 3.

However, clinical RBE modelling is dominated by two approaches, which build on microdosimetric simulations and simple assumptions about cellular fate. These are the Local Effect Model (LEM) and the Microdosimetric Kinetic Model (MKM). In both cases, the models seek to take advantage of a combination of biological response parameters determined for X-ray exposure and microdosimetric calculations to predict the effects of high LET irradiation. 

The LEM defines radiation response in terms of “lethal events”, such that survival is given by $S=e^{-N}$, where N is the number of lethal events. For X-rays, it can be seen that these lesions are formed at a rate $N=-\alpha D-\beta D^{2}$, and the LEM assumes that the same dependence holds in sub-volumes of the nucleus of cells exposed to heterogeneous radiation exposures. Thus, the total yield of lethal events across a cell is given by $N=\int \frac{-\alpha D_{r}-\beta D_{r}^{2}}{V}\;dV$, where $D_{r}$ is the dose delivered to a point r within the nucleus, and V is the total nuclear volume. It can be seen that because of the $D_{r}^{2}$ term, the total yield of lethal lesions will be greater in a heterogeneous exposure than a uniform exposure, which gives rise to elevated RBEs in such systems [102]. More recent versions of the LEM have moved to explicitly consider break repair and misrepair probabilities similar to some other models described above, but these approaches have yet to make the transition to clinical practice [103]. 

The MKM takes a conceptually similar approach. In it, the nucleus is sub-divided into a large number of small “domains”, each of which can contain either directly lethal events, or potentially lethal events. Potentially lethal events within the same sub-domain then have the chance of mis-rejoining to cause lethal events or successfully repairing, giving rise to a linear-quadratic response for lethal lesions within each micro-dosimetric domain. As in the LEM, rates for the formation of lethal and sub-lethal damages can be determined from the approximately uniform damage caused by X-ray irradiations, although the MKM uses the dose-weighted specific energy per track passing through a domain to calculate the induced damage. Because high LET exposure leads to heterogeneous cellular exposure and elevated risks of multiple sub-lethal events in a single domain, an increased probability of misrepair and thus cell death results [104].

In both cases, by combining these relatively simple biological models with microdosimetric models, good predictions can be obtained for RBEs in a range of systems. As a result, these models have successfully made the transition into clinical use, with the LEM and MKM supporting treatment planning optimization at carbon therapy institutes in Europe and Japan, respectively [105,106]. This success shows the potential of more mechanistic models to inform clinical practice and improve patient outcomes. Moreover, their mechanistic nature has enabled them to be applied and validated outside their initial area of application, with the MKM model being applied in areas such as dose rate effects in brachytherapy [107] and other modalities, such as boron neutron capture therapy [108], while the LEM has been applied to gold nanoparticle enhanced X-ray therapies [109].

However, a major challenge which is presented in this area is what events lead to cell death. For example, while the PARTRAC code is capable of simulating detailed models of initial damage and how the cell resolves it, significant uncertainties in the absolute yield of damage remain, and survival predictions still require assumptions to be made about how cells tolerate this damage [64]. Similar limitations are present in other whole-cell DNA damage codes [66].

One common approach is to assume that some subset of types of repair failure can lead to cell death. These can include combination of DSBs or other lesions into “complex” damages, which are deemed irreparable, those which are determined to be unrepaired at a particular timepoint, or those leading to significant misrepair, such as chromosome aberrations. A number of models have taken this type of approach. For example, the BIANCA [79] model simulates the interaction of “cluster lesions” in DNA to predict yields of chromosome aberrations (and later cell death) as a function of LET. The GLOBLE model [110] applies a similar concept in low LET scenarios, modelling isolated and clustered DSBs to better understand the LQ model and its temporal dependence. The “Multiscale” approach [111] also models high- and low-LET effects by simulating the formation of complex clustered lesions as the event which drives radiation-induced cell death. Finally, a combination of MCDS and RMF models [112] combines predictions of DSB yield with frequency-mean specific energy values to predict the RBE of different ions. 

While this represents only a selection of the models making use of this approach, it can be seen that this concept of grouping damages spatially into lesions which either misrepair or are deemed irreparable is very common across a wide range of models, despite substantial differences in modelled endpoint and scenarios of interest. This is likely in part motivated by the good evidence that chromosome aberrations that cause significant genetic rearrangement or loss are associated with a loss of clonal viability [72], and the good correlations which have been observed between unrepaired DSBs and cell death in some lines [71,113,114]. However, while this conceptual approach effectively captures overall trends in radiobiological responses, there remains significant disagreement between models on the exact form of this dependence, and these models have been validated to different degrees against heterogeneous datasets.

Moreover, cell death is a complex process, involving both “passive” and “active” death pathways, which can depend to varying degrees on the underlying cell biology, making it unclear by which process a given cell will actually die [115]. These pathways include apoptosis [116,117], necrosis [118], mitotic catastrophe [119], autophagy [120], and senescence [121]. Because many of these pathways require the active involvement of different genes, they can vary substantially based on both the tissue type of origin of a given cell, and any genetic alterations in the many genes involved in these processes. This can significantly impact on the overall sensitivity of different cell lines, for example, with many lymphoid cells showing dramatically higher levels of apoptosis and radiation sensitivity than cells derived from epithelial tissue or fibroblasts [122,123].

These dependencies have been extensively studied in the literature, and some published response models incorporate a high-level approximation of these pathways to reflect differences between such cell lines. However, there is again relatively limited modelling of the mechanisms of these pathways in determining radiation response. While some models do exist, they frequently focus on the pathways more generally, including seeking to understand intrinsic and signalling-driven apoptosis via chemotherapy, rather than that driven by radiation-induced damage [124,125,126,127,128]. 

Despite the significant complexity underlying these different pathways, improving our understanding of them represents an area of significant unexplored potential in our attempts to better understand radiation responses.

## 5. Tissue-Level Responses

Although this review has primarily focused on the response of individual cells, it is important to note that clinical outcomes are defined at the multi-cellular level, in terms of responses in tumors and normal tissues. While the intrinsic sensitivity of individual cells is believed to play a significant role in radiosensitivity in both preclinical and clinical models [129,130,131], other effects also potentially become significant.

While cell killing is associated with toxicity and tumor control, the precise amount of killing and the distribution throughout the target which gives rise to different endpoints is highly variable. In particular, while many tumor control probability (TCP) models assume that control depends on killing all (clonogenically viable) tumor cells, normal tissue complication probability (NTCP) can depend on the distribution of dose throughout the organ in a more complex fashion. Some organs respond to irradiation with a severity which approximately follows the average dose across the whole organ (“parallel” organs such as the lung or liver), while other organs can see significant toxicity following irradiation of only a small sub-region (“serial” organs such as the spinal cord or bowel).

A variety of modelling approaches have been proposed to analyze this problem, beginning from Poisson-based probabilistic models [132], which have been generalized using abstract approaches, such as the generalized Equivalent Uniform Dose (EUD) [133,134], which seek to incorporate these volume dependencies for different tissue types. For NTCP, two models are commonly used. The first is the Lyman-Kutcher-Burman (LKB) model, which applies a tissue-specific empirical volume scaling to correct for irradiation of different sub-volumes to different doses, reflecting the difference between serial and parallel organs [135,136]. An alternative process, which seeks to incorporate knowledge of the structure of different tissue types, is the Relative Seriality model [137]. This model described organs in terms of their “seriality”, with highly serial organs sensitive to toxicity arising due to damage in very small volumes. However, as both of these approaches tend to abstract out significant amounts of mechanistic information, there is also interest in more mechanistic approaches, such as tumor models, which incorporate more details of proliferation, heterogeneity, and hypoxia [138,139], and normal tissue models, which more explicitly define “functional sub-units” of the organs and how they interrelate to incorporate these structural effects [140]. 

A final tissue-level endpoint of interest is carcinogenesis. While the majority of predictions of cancer risk following radiation exposure are statistical, based on extrapolation from populations such as the atomic bomb cohorts, the significant uncertainty in cancer risk at low doses means there is considerable interest in using more mechanistic models to mitigate this uncertainty. Some of these models remain stochastic, while incorporating more information on the stages of carcinogenesis, while others seek to develop more comprehensive models from the cell level, simulating DNA repair failure and mutation, together with clonal expansion driving cancer formation [141,142,143,144]. 

However, in both tumor and normal tissue models, there remains significant uncertainty in model parameters, and there is typically limited incorporation of information from the single-cell level beyond overall radiosensitivity, suggesting a further challenge in translating from cell-level models to clinical responses.

In addition, irradiation in tissues opens the possibility of effects resulting from intercellular communication. The two major pathways of this effect are the radiation induced bystander effect (RIBE), and immune effects. In the RIBE, signals from neighboring cells can cause unirradiated cells to suffer genomic stress and potentially die [145]. This is a departure from traditional radiobiology, where radiation responses are predicted based on the dose delivered to the irradiated cell alone. A number of RIBE models have been published, but are typically not incorporated in clinical planning [146,147,148]. The involvement of the immune system in radiotherapy is an area of rapidly growing interest, with substantial evidence that changes in regulation of immune signals can play a significant role in determining tumor control following radiotherapy [149,150]. As with the RIBE, a number of models of radio-immunotherapy combinations have been published [151], but as this is a new and rapidly developing field, once again they have seen little translation to clinical practice.

The step between cell-based models and tissue-level responses remains a major challenge in radiobiological modelling. Tissue structure introduces the possibility for complex volume effects which are not apparent on the single-cell level, and this significant increase in complexity is combined with limited experimental data, presenting a substantial challenge for modelers to develop useful and robustly validated tools to link our extensive preclinical knowledge to clinically-meaningful predictions.

## 6. Potential Impacts of Modelling Advances

A recurring theme in many of the above discussions is the substantial biological complexity that underlies many aspects of radiation response, which is obscured by the relatively simple response models that can successfully characterize both pre-clinical and clinical responses. In many cases, it is reasonable to question how much benefit will be derived from the development of these much more sophisticated models, or if relatively simple, well-established approaches may be sufficient to describe all aspects of radiation response.

However, there are a number of major outstanding clinical questions which these simpler models are not suited to address. One of the most significant of these is that there is now extensive clinical evidence that even in a group of patients with tumors of the same type, there may be a broad range of radiation sensitivities among their individual tumors [132], driven in large part by the cancers’ spectrum of underlying mutations [25]. Estimates of this variation from clinical response curves suggest it could represent large differences in radiation sensitivity even for cancers which would be treated identically in the clinic (e.g., variations in 25% or more in the α component [132]). Consequently, a large fraction of patients are almost certainly being under- and over-dosed by these population-based models, which may significantly impact on clinical outcomes. Empirical population-level models offer no way to address these issues and predict the sensitivity of an individual cancer, as their model parameters are typically at least somewhat abstract and cannot be linked directly to particular genetic pathways or tumor characteristics. This presents a significant challenge in radiation therapy, preventing it from becoming a fully personalized therapy.

Similar issues arise in a number of other areas which can impact on tumor response—the effects of factors such as the tumor microenvironment, hypoxia, and radiation quality are not easily incorporated in these models, at best being addressed by empirical fitting parameters such as OER and RBE effects. These represent significant additional avenues by which treatments could be personalized, if they could be more robustly incorporated in biological models of tissue and tumor response.

As a result, there has been significant interest in the development of clinical signatures of radiation sensitivity. It has been reported that the in vitro clonogenic radiation sensitivity of cells cultured from tumors correlates well with tumor sensitivity in both animal models [130,152] and patients [153,154,155], along with some other intermediate endpoints, such as residual DNA damage in ex vivo irradiated samples. However, such assays are often labor-intensive and difficult to incorporate into a clinical workflow [156]. More recently, there has been a dramatic increase in interest in the use of genetic signatures of radiation response. These techniques seek to associate alterations in the mutation status or expression of genes with radiation sensitivity, typically using purely statistical association. A number of signatures have been reported, either focusing on radiation response in general [157,158], or specific modulators of response, such as signatures of hypoxia [159]. While a promising approach, many of these signatures have faced challenges around reproducibility and robustness [160], as well as questions about if they are truly predictive of radiation response or merely more generally prognostic [161]. In addition, the volume of data required to fully parameterize such empirical models is often prohibitive—as an example, there is still little agreement on the true magnitude of the RBE of protons, despite several hundred studies reporting results in this area [162].

The incorporation of more detailed mechanistic models has the potential to help address these challenges. Models that enable the integration of statistical, population-level data with our growing mechanistic knowledge of the underlying drivers of radiation response can enable the generation of more robust and translatable predictions. Some models have demonstrated the basic potential for predicting aspects of radiation response based on phenotypic or genotypic characteristics, although this remains a significant area of ongoing research.

Such models can potentially be useful in other areas also. For example, radiation is rarely delivered as a single agent in modern clinical practice, instead being combined with chemotherapy or new targeted therapies to improve outcome. While these are typically prescribed based on their activity as single agents, there is growing interest in specifically seeking to develop radiation sensitizing agents [23]. In this approach, agents are usually identified which seek to target specific parts of the radiation response pathway, such as genes involved in DNA repair [163,164,165]. While this approach has identified some radiation sensitizers, translation has been limited. More robust mechanistic models may enable better predictions of the impact of modifying these aspects of DNA repair without the need for extensive preclinical testing.

However, in all cases, it is important to ensure that model development is carried out in a rational way. As noted in the discussion of alternative clonogenic survival models in Section 1 and DNA damage models in Section 2, as model complexity increases, it can become increasingly challenging to distinguish between different models due to a proliferation of adjustable parameters, and added complexity may actually impact negatively on model predictive power [166]. Thus, it is important to ensure that that any additional modelling complexity adds value—either by enabling explanation of novel phenomena, or by enabling additional data sources to be exploited to refine and constrain model predictions—and ideally, be feasible to test and validate in a clinical setting. 

## 7. Conclusions

Mathematical modelling has been a key part of radiation biology and therapy for almost a century, and continues to play an important role in research developments. While a number of relatively simple and largely empirical models dominate current clinical and pre-clinical practice, significant advances are being made in our understanding of all aspects of radiation response, including the initial physics, resulting chemistry, and subsequent biological consequences. Although this is a very complex and challenging area of research, applied correctly it has significant potential to advance our understanding of the drivers of radiation response, and potentially translate this into useful clinical predictive tools in the future, if strong collaboration can be maintained between the diverse disciplines in radiation research.

## Figures and Tables

**Figure 1 cancers-11-00205-f001:**
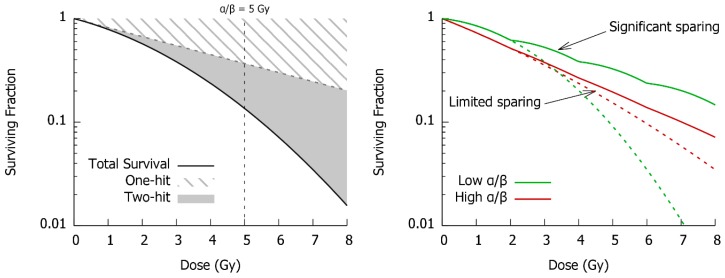
Left: Illustration of linear-quadratic dose response model. This postulates that cell survival has a “one-hit” linear component, and a “two-hit” quadratic component, which contribute varying amounts with varying doses. While simple, this model has been shown to encapsulate many aspects of radiation responses. Right: As an illustration, it provides an intuitive explanation for the benefits of fractionation, where in low α/β tissues the “shoulder” is repeated, giving rise to significant sparing, while in high α/β tissues, much less sparing is seen. However, this model has only limited scope for mechanistic applications.

**Figure 2 cancers-11-00205-f002:**
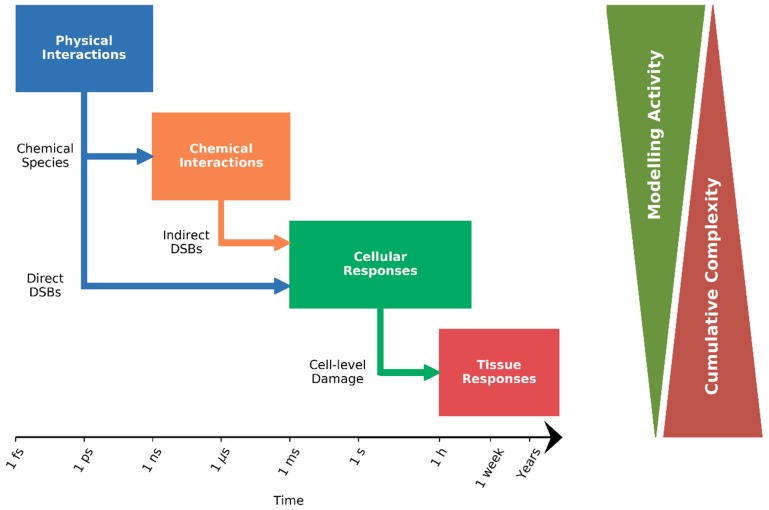
Illustration of stages of a comprehensive mechanistic response model, and approximate associated timescale. Early responses are dominated by physical interactions, models of which generate distributions of reactive chemical species, and direct DSBs generated in typically less than a nanosecond. These chemical species may also be included in a chemical interaction model, which generates predictions of indirect DSBs, over a scale of 1 ms or less. Responses at the cell level then take place over hours to days, as cells repair this damage and potentially respond by arresting, dying, or seeing other changes in function. Finally, tissue-level responses integrate such responses over a scale of hours to years, giving rise to clinically observable effects. Mechanistic modelling activity in radiotherapy has primarily focused early in this timescale, on physico-chemical and early cellular responses, with relatively little modelling of whole-tissue responses. This is in part due to the significant cumulative complexity in describing all aspects of these responses in a single system.

**Figure 3 cancers-11-00205-f003:**
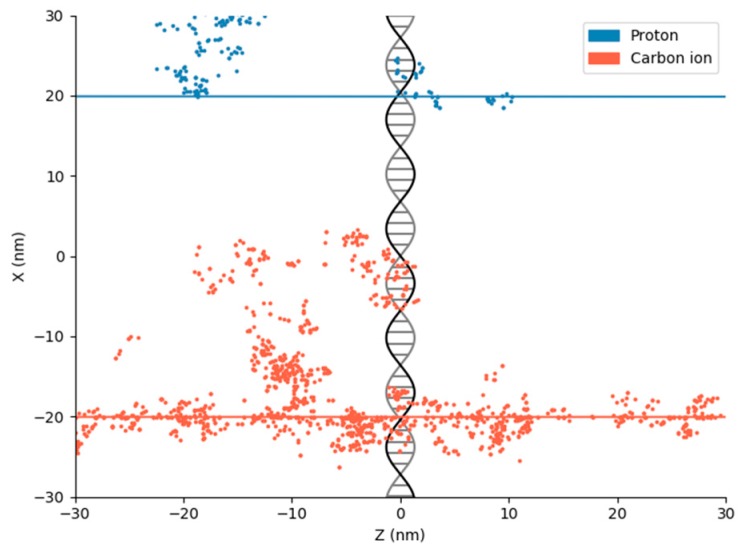
Illustration of track structure and energy deposition distributions for a 10 MeV proton (~5 keV/µm LET, blue, top) and 200 MeV carbon ion (~100 keV/µm LET, red, bottom), together with a schematic representation of DNA for scale. For both particles, the primary particle track (lines) deposits only a small fraction of the total energy, with large amounts of energy being deposited in the vicinity of the track core by low energy secondary electrons, with individual energy depositions here shown as points. It can be seen that while the proton deposits energy relatively sparsely, the carbon ion deposits energy extremely densely along its track, potentially causing multiple closely spaced damages within any DNA it traverses.

**Figure 4 cancers-11-00205-f004:**
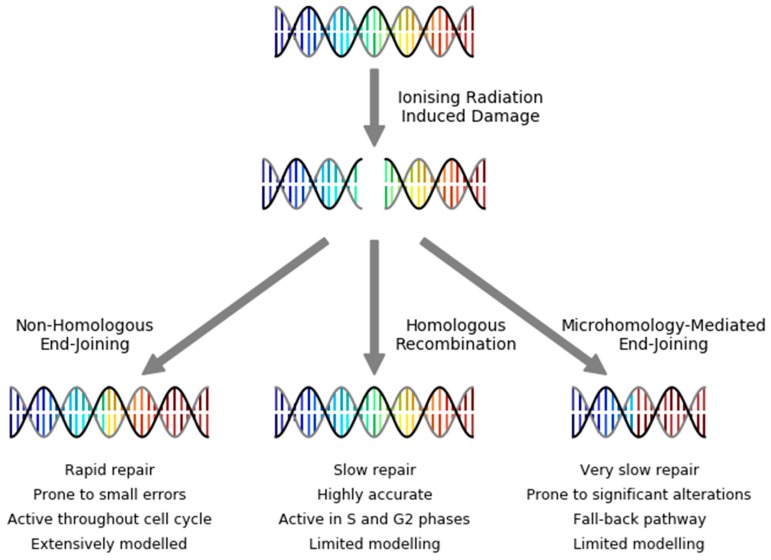
Illustration of double-strand break repair pathways, with DNA sequence progression schematically illustrated by color scale, to highlight sequence alterations. Ionizing radiation induced double strand breaks (top) can be repaired by one of three pathways—Non-Homologous End Joining (NHEJ, left), Homologous Recombination (HR, center), or Microhomology-Mediated End Joining (MMEJ, right). These pathways differ in their efficiency, fidelity, and activity throughout the cell cycle. NHEJ has been the subject of extensive mechanistic modelling in radiotherapy, while relatively few studies have been published of the other pathways.

**Table 1 cancers-11-00205-t001:** List of Monte Carlo models of DNA damage, which combine various track structure physics codes with a variety of DNA models to generate predictions of different classes of DNA damage. A broad range of underlying codes and models have been used to simulate damage from a range of radiation types.

Paper	Physics Code	DNA Model	Endpoints
Nikjoo et al. [39,63]	KURBUC/PITS	Whole nucleus containing chromatin fibers arranged in hierarchy of spherical volumes	DSBs, SSBs, base damages
Friedland et al. [44,64]	PARTRAC	Whole nucleus containing model chromatin fiber random walk	DSBs, SSBs, DNA fragment sizes
Bernal et al. [65]	Geant4-DNA	Atomistic DNA segment model in cube	DSBs, SSBs
Plante et al. [41]	RITRACKS	Flexible polymer chain chromosome model in nucleus	DSBs
Meylan et al. [66]	Geant4-DNA	DNAFabric-based nucleus model	DSBs, SSBs
McNamara et al. [51,57]	Geant4-DNA/TOPAS-nBio	Multiple DNA structures, fractal walk nucleus	DSBs, SSBs

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
