# Peer review of "Mechanistic Modelling of Radiation Responses"

_cancers, 2019, doi:10.3390/cancers11020205_

Round 1
Reviewer 1 Report
The authors McMahon and Prise provide a nice and condensed review, attempting to summarize mathematical modelling approaches in the field of radiation biology and radiation therapy. Their goal was to set an emphasis on current modelling activities, highlighting the perspectives promised by such research.
In my opinion, the manuscript is an helpful overview for both researchers involved in modelling and oncologists or biologists which might take advantage from existing model work or gain more insight in such. The manuscript is well organized, being oriented along the cascade of radiation damage from physical down to the medical stage over orders of spatial and temporal magnitudes. In particular by that choice of organization the authors succeeded to include the wide span of existing model approaches for different endpoints, different levels of detail and model techniques in a condensed text.
At some points, however, to my feeling there is considerable imbalance in the presentation of different models concerning their levels of validation and applicability. Also once in a while some more precision and transparency concerning the basic ideas underlying the models is needed. Finally, the future perspective of integrating more experimental knowledge into model work might be debated and thus should be presented in a more moderate manner.
I shall outline this criticism in detail below, followed by many minor comments which hopefully will be helpful to the authors for revising their manuscript.
1.) Imbalance in model presentation: Many models have been developed throughout the history of radiobiology, but many of them have been applied to one specific scenario (e.g. a few dose response curves), and have not followed any more. Others have been used extensively and confirmed by application to multiple endpoints or phenomena. Here the state of the art should be appreciated, without assessing the quality of the models. Authors should ask the following questions:
- To what extent have the various model approaches been applied to real-world situations?
- To what extent have they been validated?
- To what extent have model ideas proven as comprehensive by applying the model to scenarios it was not developed for originally?
To give some examples, some models of relevance are not mentioned at all explicitly (such as RMR and RMF and saturated repair models) and therefore should be added with a brief discussion. Moreover, the models triggered further research to different levels, in that regard in particular the wide range of research exploiting the LQ model, the LEM and MKM should be indicated. Finally, for the sake of completeness some words about mechanistic modelling of radiation induced carcinogenesis should be added, as in particular this field - while traditionally being covered mostly in epidemiology - is more and more impacted by radiobiology research.
2.) Lack of precision: Some remarks about the referenced models are not entirely correct or at least could be critically discussed. I marked these points in the detailed comments below. As a part of geving credits to models and associated findings in an appropriate way I also suggest to consequently cite the original papers as references. This point is important in particular in a review.
3.) Conclusion: Essentially the authors say that the implementation of gain in radiobiology knowledge will lead to more integrative models. However, I do not see a straightforward argument why this is a promising vision, as a higher level of detail and model complexity not necessarily goes along with a better accuracy of model predictions. Indeed, it is well known that a too large number of free model parameters makes parameters less determinable from experimental data and therefore models less robust, see e.g. van der Schaaf, IJROBP 2015. So, to my opinion we have to question primarily what type of modelling (mechanistic / empirical, level of accuracy, ...) is needed, what the most relevant processes are to be considered in modelling, and what open biological / medical questions are tempting to approach by the mutual contribution of experimental and modelling research. Maybe these thoughts inspire the authors to adjust their final comments towards a more open outlook.
Remarks on the scientific content:
----------------------------------
p2, Fig 1: (i) There is no difference in the meaning of the "alpha component" and the "one-hit" shaded area, it's just two ways of expressing the same - should thus be condensed. (ii) The meaning of the a/b ratio (here chosen as 5 Gy) of having equal contribution of both components should be appreciated in the text or figure caption.
p2, l49-51 and 70: It should be recognized that the LQ model was used successfully previously in the context of cytogenetics (since Lea [Catcheside, J Genetics 1946]). Sinclair and Fowler started to use it for the endpoint of cell survival [Sinclair, IAEA Tech Rep 58, 1966; Douglas & Fowler Radiat Res 1972]. This double use lead to immediate mechanistic interpretations using the idea of binary misrepair by erroneous rejoining of DSB. The idea finally got manifested first in the RMR model (please adjust ref [11] as the original publication of the RMR was in 1980 already [Tobias, in: Radiation biology in cancer research, Raven press, 1980]) and later in the LPL.
p2 l 63-64: While the previous part of the paragraph regards in-vitro cell survival, here the a/b ratio is used in a clinical context, where it is usually derived from a comparison of different fractionation schemes. This difference should be emphasized briefly to avoid solely interpreting the a/b ratio as a characteristic quantity of cell inactivation.
p2, l 67-68 and Fig 1: The terminology of "tracks" and "hits" is rather misleading, in particular in the case of photon radiation (what would be a track or a hit there?). I rather suggest to talk about the interaction of portions of radiation (which is why we have a dose^2 dependence lead by the beta coefficient). In the case of particle radiation this reduces to interaction between particle tracks, as within one track all energy is deposited in a correlated way (thus leading to an enhancement in the alpha term). The idea that the beta term indicates the strength of radiation interaction with respect to the effect is still the most basic and broadly accepted interpretation of the LQ, which could be remarked after the discussion of diverse interpretations ending in line 74.
p2, l 75: "...it has proven challenging to make the LQ model mechanically robust...": This is at least highly debatable, and there are many researchers who tend to refute the mechanistic underpinning of the LQ. A particular example where the LQ model in its simplicity brings us quite far is the consistent understanding of dose rate effects, that effectively lowers the beta term for protracted irradiation.
p3, Fig 2: Consider to include also larger time scales than weeks: Normal tissue reactions range into years, e.g. radiation induced second cancers.
p3, l 77-79: A reference to the 4 R's of radiobiology as modifiers of radio response and how they have been attempted into LQ oriented approaches (see various work of D. Brenner) would be helpful.
p3, l 79-81: Why does the LQ model neglect the formation of potentially carcinogenic mutations? First, the LQ model is devoted to different endpoints than carcinogenic lesions, and second, concerning neoplastic transformations, which may be regarded as a first step on the road to cancer, the LQ model has been successfully applied to in-vitro assays (see works of Miller and Young, e.g.).
p3, l 87: "...there is not as yet a single integrated model of these processes": This is way too pessimistic. Several models combine phenomena and processes on different levels. E.g. PARTRAC, LEM and RMF and the multiscale approach all cover DSB induction and repair, which take place on different scales. Considering the needed compromise of model complexity and meaningful model parameters no model will ever reflect all processes (which are even not fully known) in any detail. Hence, the question is again what the purpose of modelling is and what the relevant processes to be included are (see point 3 above).
p4, l 94: There is the seminal work of Munro, who irradiated cell nuclei and cytoplasm selectively with an alpha pinpoint irradiator. I think it's appropriate to give credits to this work [Munro, Radiat Res 1970].
p4, l 99: "frequently quoted over relatively large volumes": Following ICRU 60 dose would be defined as mean value in arbitrary small volumes. Hence dose is a point like mean (enemble average) over energy deposit patterns rather than a mean over a macroscopic volume. If in macroscopic volumes there are dose gradients, more elaborate techniques (such as DVH or other dose reduction methods) could be used.
p4, l 101: "two turns": This sounds like a hard criterion, which it probably is not.
p4, l107-112: While the rest of the paragraph only deals with DSB induction, here finally the RBE is introduced, while in the following section a step back is made towards physics simulation. Consider to introduce RBE later in the manuscript. Moreover the definition of RBE is flipped (RBE=photon dose / ion dose, not ion dose / photon dose).
p4, l 113: I would strongly object that Monte Carlo techniques provide a "gold standard". For reasons discussed later in the manuscript the MC methods are just as accurate as the cut off energies and input cross sections allow them to be. There is no direct advantage of MC calculations that do the physics in great detail while the conversion of energy deposits into DSB or other DNA lesions depends on assumptions just as non-MC models do. The gold standard would rather be consistent sets of experiments which allowed testing such assumptions and models built upon.
p4, l 119: "...heterogeneity which can be seen even in cells...": This formulation might be misleading: Actually, if for example a dose of 1 Gy photons is given to nuclei of same size, the deposited energy in these nuclei varies only by a small factor. Hence there is only small heterogeneity in cells. Only within smaller volumes like microdosimetry sites there is considerable heterogeneity (which the authors probably meant).
p4, Fig 3: What are the dots, ionizations or both ionizations and excitations? What is meant in the caption by "multiple overlapping damages", considering that energy transfer is local (point like) within quantum mechanical uncertainty?
p4, l 94-97: This sentence could suggest that DSB are lesions with a high lethality probability for the cell, which is not the case at all, as most DSB will be repaired successfully (as noted later in the manuscript). It would be instructive to give some numbers of induced DSB per Gy already here. In a revised worded interpretation, only more complex subclasses of DSB are thought to be most relevant for triggering cell inactivation.
p4, l 106: "amounts" --> "amounts per Gy" or "amounts per unit dose"
p5, l 125: What is meant by "By fitting the sensitivity of various parts of the DNA to ionizing events"? What quantity?
p5, l 128/129: Complex multiply damaged sites are not (yet) a directly measurable quantity, rather they are proposed to be a highly relevant class of lesions.
p5, l 140: What about PARTRAC as one of the most elaborate codes concerning the biology application?
p5, l 140 and Tab 1. A brief summary of the achievements and fields of application of the different codes would be helpful. E.g., KURBUC has been used to study the impact of electron track ends, PARTRAC for investigating mechanisms leading to chromosome aberrations, and Geant4-bio is much exploited in the nanodosimetry community, where the accumulation of energy in nanodosimetric volume is set in relation to provoked biologic effects. (See also my comment 1 above)
p5, l 144: The last point in the list (understanding the events that drive DSB) should be more emphasized: This is where physics meets biology, and having not fully understood the underlying processes this is the point where most uncertainty is emerging.
p5, l 152: Such research was already extensively pushed forward in the 80s and 90s by Goodhead and Nikjoo, maybe it is appropriate include corresponding references.
p5, l 154: "30 %": Please include a remark that this number refers to high energetic photon radiation.
p5, l 161: Another dedicated code is TRAX-CHEM (Boscolo, Chem Phys Lett 2018).
p6, 1st para: In addition to the unknown energies needed for single strand breaks, also experimental methods to evaluate DNA damage are subject to uncertainty. So it's not only the underlying theory and physical cross section but also the quality of existing biology data that contribute to the problem.
p6, l 213/214: Why do models that evaluate damage yields down to the level of chromosome aberrations have limited capacity of incorporating more knowledge of DNA repair? In the course of transforming induced DNA lesions into final aberrations it should be feasible to include such repair aspects, in principle. However, there are general shortcomings in this class of models as there seems to be no direct correspondence between appropriate subsets of chromosomal aberrations and cell inactivation (there are correlations, but e.g. a/b ratios for asymmetric aberrations do hardly correspond to a/b ratios for survival).
p7, l 217 and Fig 4: The picture and text suggest that there is a unique alternative pathway. However, this is matter of ongoing research, where several mechanisms have been studied that seem to have substantial differences e.g. concerning the involved ligases, see e.g. [Löbrich & Jeggo, Trends in Biochem Sciences 2017; Iliakis Radiother Oncol 2009]. Hence presumably we have multiple alternative pathways, the importance and employment triggers of which is still somewhat unclear. Likewise, their speed is not necessarily "very slow".
p8, l230-235: Actually, repair kinetics is an often studied endpoint for which also a wide span of modelling approaches exists. It should be mentioned that in addition to full pathway modelling also empirical models like the biphasic double exponential approach exist. Moreover, the attribution of the fast and slow phase to pathways might not be the only interpretation of observed repair kinetics.
p8, l 243: "...robust descriptions of these pathways are needed to accurately predict the consequences of repair." This seems not convincing for the following reason: When the success rate of the available pathways to a specific kind of lesion is known, or when the overall success rate of all pathways is known (empirically), this is sufficient for effect predictions as well, even without detailed mechanistic insights. Again, the choice of model approach depends on the problem to be tackled, and we have to answer the question what level of mechanistic detail will be beneficial/needed for the approached task (see also general comment 3).
p8, l 246: Pathway choice may not only depend on lesion type, but also on lesion frequency (and therefore on dose).
p8, l 261 - 263: Both by historical importance and further applications the Repair-Misrepair model by Tobias should also be mentioned here.
p8, last para: It seems a bit uncommon and out of the general systematics of the manuscript to start with "empiric only" models for the specific case of protons here. Better bring the special case after mechanistic RBE models.
p9, l 278: In contrast to the more mechanistic models one should point out that RBE dependence on dose is neglected in the empiric approach.
p9, l 279: While MKM and LEM are most used, the Katz model [Butts & Katz Radiat Res 1967] and the Repair-misrepair-Fixation model [Carlson Radiat Res 2008] should at least be mentioned. The latter one has been applied to a plethora of phenomena, including different endpoints, although not (yet) used in therapy planning.
p9, l 300: It would be instructive to briefly summarize the fields of applications and extension of the MKM and LEM models to demonstrate their range of benchmarking and application. Among other aspects, the MKM has been applied to understand dose rate effects in brachytherapy and other radiation scenarios like the application in boron neutron capture therapy, and LEM has been used for modelling the effects of soft X-rays and kinetic aspects of repair, carcinogenesis and sensitization effect of nanoparticles.
p9, l 284: Concerning LEM, the presented idea follows an older version, while in the current LEM model simulation of DSB pattern according to local dose is an integrated model part. While the newer version allows a more mechanistic view, the original model is still applied in treatment planning in the clinics.
p9, l 292: Concerning the MKM, it is an extension to Kellerer's theory, and besides the notion of sublethality the main idea was the linear-quadratic response of any microdosimetric domain.
p9, l 303: "Germany" --> "Europe" (Also in Italy and Austria that biological model is used in treatment planning).
p9, l 306-309: As in any model, in particular the initial damage depends on underlying assumptions in PARTRAC. Hence "detailed" should not mean "precise" or "with small uncertainty", maybe another wording could make this point clearer.
p9, l314-315: The list of these models joins very different models regarding techniques and purposes. Also the idea of complexity which is their joint feature differs. Here a more distinguished consideration would be needed: The BIANCA model is a high LET model based on chromosome aberration formation facilitated by means of a LET dependent fit parameter; GLOBLE as a pure low LET model (such as LQ) with a discrimination of two damage classes of different severity, the multiscale approach is another Monte Carlo transport approach for ion radiation including also condensed matter effects (Coulomb explosion), and the MCDS is a stochastic model for assessing SSB and DSB induction in different complexity variants (not a model for cell survival at all). Likewise, again different levels of model applications and benchmarking would be appreciated. (See comment 1).
p9, l 317-318: "and good correlations have been observed between unrepaired DSBs and cell death in some lines." This is certainly true, but often misunderstood: The correlations have been found on a double log scale, but there is no direct correspondence of, say the alpha/beta value of asymmetric chromosome aberrations and cell inactivation.
p10, l 327: Please provide a reference for that statement.
p10, l 356: In addition to the more phenomenological Lyman-Kutcher-Burman model, the relative seriality model of Kallmann and Brahme allows a more mechanistic interpretation and has found considerable use in the literature. It would be adequate to spend a sentence on it as well [Kallmann & Brahme IJRB 1992].
p10, l363-365: This is rather true for NTCP than for TCP, as for the latter a simple Poisson (maybe enhanced with cell repopulation terms) approach already results in a reasonable model.
p10, l 375: Reference [120] does not cover combined response including radiation, would drop this reference.
p11, 377-388: One of the biggest challenges here is a proper understanding and implementation of volume effects of partial organ irradiation. As this triggered many experimental studies and several quantities like EUD etc. a further note would be helpful.
p11, 393: To what measure for radiosensitivity refer the 25%?
p11, l 408/409: For a review it would be more appropriate to provide original work as references. Concerning the difference of in vitro and in vivo radiosensitivity, major steps have been initially taken by Fertil & Malaise [IJROBP 7, 621], Bristow [IJROBP 18, 133] and Kelland et al. [Radiother Oncol 16, 55].
p11, l 424: This statement is somewhat vague, as another question has to be considered in that context is: What modelling results are actually needed, where is improvement requested most? See comment 3 above.
p12, l 339-441: This is an always true statement - but again the question would be to clearly define the goals of current and future modeling: What is the potentially "added" benefits? See also comment 3 above.
Suggestions on wording etc.:
----------------------------
p1, l 34: "These models dominated studies on bacteria..." Rather experiments triggered modelling, not vice versa.
p2, l 53: The LQ is rather a simple model or formula, not a "technique".
p3, Fig2: Axes labels: (i) us --> µs, (ii) hr -> h
p3, l 84-85: "the cell's fate hours" --> "the cell's fate within hours"
p4, Fig 3: In the caption "can be see" --> "can be seen"
p6, l 177: "models been" --> "models have been"
p8, l 253-255: "Improved...response" Such a statement is always true in modelling and just recalls the scientific method. No need to list it here.
p8, l 260: Expression "high-level" is somewhat undefined, does it refer to level of detail, mathematical complexness, performance or accuracy?
p9, l 302: "have been successfully made" --> "have successfully made"
p9, l 321: "given will actually" --> "given cell will actually"
p9, l 322/323: Inflation of "significantly"
p10, l 346: "In particular, many tumour..." --> "In particular, while many tumour..."
Reviewer 2 Report
A very comprehensive review that outlines various modelling approaches on different scales to study the effects and responses of radiation.
Reviewer 3 Report
The manuscript presents a review of mechanistic modelling of radiation responses. It describes models dedicated to capture the initial DNA damage induction, its cellular repair, the link between residual damage and cell killing, as well as tissue-level responses. The paper is in general well balanced, well structured, clearly written, and leaves very little to be desired.
General comments:
Consider mentioning also mechanistic models of radiation responses in radiation protection, in particular models of carcinogenesis (recently reviewed by Rühm et al. Int. J. Radiat. Biol. 2017;93:1093-1117). Also mechanistic models of radiotherapy-induced second cancers or cardiovascular diseases have been delevoped (Shuryak et al. Radiat. Environ. Biophys. 2009;48:275-286, Schneider Med. Phys. 2009;36:1138-1143).
To this reviewer, the major issue hampering the application of mechanistic models in clinical radiobiology and radiotherapy (pp.11-12) is the difficulty in transferring in vitro results to in vivo conditions. Clinical data are typically sufficient to derive a few (2-3) parameters only, but lack the statistical power needed to derive the number of parameters included in mechanistic models. From this perspective, for instance the last paragraph of p.11 appears a bit overly optimistic. Even if robust mechanistic models were available, their implications would still need to be preclinically and clinically validated (l.436).
Specific comments:
L.32: consider briefly explaining why an exponential dose-dependence results
Fig.3: Consider including photon and/or electron tracks for comparison. How were the tracks generated? Fig. caption l.1: 'ionisation', l.6 'It can be seen that ...'
L.99: 'but this does not ...'
L.104: 'charged particles' -> 'high-LET particles'?
LL.125, 140: studies by Paretzke, Friedland et al. deserve mentioning here too
L.154: 'is' -> 'are'
L.161: Modelling chemical species & their reactions is included in PARTRAC, cf. Ref. 43. L.164: DNA structures are ignored when generating physical tracks, but considered explicitly in the subsequent chemical stage.
L.168: Ref.46 describes a quick procedure to re-calculate DNA damage data from full track structure simulations, and as such does not really fit in here
Table 1: The description of DNA models is not specific enough, for instance the one for KURBUC applies to PARTRAC as well. Likewise, break complexity is also accounted for in KURBUC & PARTRAC. Do all codes include both direct and indirect effects? L.180: Presumably, Ref.48 includes direct effects only.
L.177: 'been' -> 'have been'
L.230: extra 'fully' or 'full'?
L.236: especially if limited in their number, these models shall be cited
L.255: 'better characterise'
L.278: consider replacing 'heavy' -> 'heavier', since carbon ions are often called light ions
L.294: 'rejoining' -> 'misrejoining'?
L.302: extra 'been'
LL.322-3: twice 'significantly'
L.340: 'at the level'
L.347: 'while NTCP ...'
L.390: extra 'evidence'; 'radiations' -> 'radiation'
LL.391, 394: Is Ref.105 on inhomogeneities appropriate here?
Please check the references for formatting etc.: capitalizing; ll.470, 548 capitalize initials, l.490 'DNA', ll. 541, 640 correct '??', ll.681-2 paper title, l. 682 'prediction'.
Reviewer 4 Report
This paper presents an overview of the impact of mathematical models in radiobiology and radiotherapy outcomes and their present and future role in clinical practice. The subject of the paper is both import and of interest to the radiation research community and it certainly worth publishing in this journal. Although the paper is well-written, well-organized and focused, below are a few comments for further improving this already good paper.
Line 98: “is frequently quoted averaged” check the English
Lines 103-106: This sentence identifies X-rays as sparely ionizing radiation and charged particles as densely ionizing radiation which is not correct in general (depends on energy, e.g., soft X-rays may be more densely ionizing compared to high-energy charged particles). I suggest to either stick to the standard low- versus high-LET distinction (not entirely correct but serves your purpose) or to be more specific, e.g., replace X-rays” by “high-energy X-rays” and “charged particles” by “heavy charged particles” or “low-energy charged particles”
Lines 107-110: Revisit the definition of RBE (RBE=sparsely (ref)/densely(test))
Lines 130-140: The present functionalities of Geant4-DNA in relation to track structure calculations have been recently reviewed by Incerti et al. Med. Phys. 45, 722-739, 2018. Also, in this paragraph, a distinction must be made between macroscopic Monte Carlo codes (e.g., MCNP, EGS, FLUKA) that employ condensed-history physics models, which are the gold-standard for patient dosimetry, and microscopic Monte Carlo codes (the rest of the codes cited) that employ track-structure physics models which are the gold standard for cellular dosimetry (refer to Kyriakou et al. J Appl. Phys. 122, 024303, 2017; and Lazarakis et al. Biom. Phys. Eng. Express 4, 024001, 2018 for a discussion and comparison between these models). As written, all codes are put in the same basket.
Along with (or instead of) ref. 30, you must also cite the more recent review paper Bernal et al. Physica Medica 31, 861-874, 2015.
Lines 153-155: The 30% direct contribution may be correct for aqueous solutions (still not sure) but not for cells (and certainly not well-established). The general notion is that it varies between 30-70% depending on radiation quality and many other factors.
Line 180: Along with ref. (46, 48, 50), the recent paper by Lampe et al. (Physica Medica 48, 135-145, 2018) must also be cited since it provides mechanistic DNA damage simulations (using the most recent models of Geant4-DNA) and offers a quantitative assessment of the role of some simulation parameters.
Table 1: Along with ref. 132 add ref. 36 in Nikjoo et al.
Line 236: “By contrast” implies that the preceding paragraphs refer to NHEJ, which is not at all clear.
Line 274: must read “where alpha_x is the” (the subscript x is missing from alpha)
Line 279: Perhaps better “In this context”
Lines 292-299: The discussion on the MKM is somewhat misleading in the sense that, contrary to the LEM, it actually considers the stochastic variation of energy deposition in submicron domains through the dose-weighted specific energy for single-tracks (z1_D).
Line 321: “a given will actually die” (something is missing)
Line 390: delete “evidence” (it appears twice)
Line 391: It is not entirely clear to me what “individual tumor types” means. Do you mean that the sensitivity varies across tumors of the same type? Clarify.
Line 443: Replace “Radiobiological modelling” by “Mathematical modelling” for consistency with the rest of the paper.
Round 2
Reviewer 1 Report
The authors improved their review manuscript significantly, as many aspects are now presented in a more balanced way. To my opinion the manuscript is suitable for publication. One point which I feel could still be a source of misunderstanding is the close and barely commented clustering of many quite distinct models (BIANCA...MCDS), where brief comments on underlying ideas and achieved validation would raise precision and appreciate the broadness of multiple approaches that have been developed for the same goal.
Author Response
As recommended by the reviewer, we have expanded the referenced section of text (beginning around line 417) to specifically identify the main endpoints of each model, their basic assumption and primary field of application to highlight the range of different approaches to which the underlying model of complex lesions leading to un- and mis-repaired damage have been applied.
The rest of the paragraph has been updated to highlight the degree of disagreement in the exact nature of the model, and the corresponding variability in validation.
As before, new and edited text is highlighted in red.